Modular automatic design of collective behaviors for robots endowed with local communication capabilities

Hasselmann Ken
Birattari Mauro mbiro@ulb.ac.be
IRIDIA, Université Libre de Bruxelles , Brussels , Belgium
Galán José Manuel
Electronic publication date: 2020 Aug 17
Publication date: 2020
Volume: 6
Electronic Location ID: e291
Received 2020 May 13; Accepted 2020 Jul 20
Copyright: ©2020 Hasselmann and Birattari
Copyright year: 2020
Copyright holder: Hasselmann and Birattari
License: This is an open access article distributed under the terms of the Creative Commons Attribution License, which permits unrestricted use, distribution, reproduction and adaptation in any medium and for any purpose provided that it is properly attributed. For attribution, the original author(s), title, publication source (PeerJ Computer Science) and either DOI or URL of the article must be cited.
License URL: https://creativecommons.org/licenses/by/4.0/

Keywords: Swarm robotics, Communication, Automatic design, Swarm, Robotics

Funding: The European Union’s Horizon 2020 research and innovation programme 681872 Belgian Fonds de la Recherche Scientifique–FNRS The research has received funding from the European Research Council (ERC) under the European Union’s Horizon 2020 research and innovation programme (grant agreement No 681872). Mauro Birattari received support from the Belgian Fonds de la Recherche Scientifique–FNRS. The funders had no role in study design, data collection and analysis, decision to publish, or preparation of the manuscript.

==============================
We investigate the automatic design of communication in swarm robotics through two studies. We first introduce Gianduja an automatic design method that generates collective behaviors for robot swarms in which individuals can locally exchange a message whose semantics is not a priori fixed. It is the automatic design process that, on a per-mission basis, defines the conditions under which the message is sent and the effect that it has on the receiving peers. Then, we extend Gianduja to Gianduja2 and Gianduja 3, which target robots that can exchange multiple distinct messages. Also in this case, the semantics of the messages is automatically defined on a per-mission basis by the design process. Gianduja and its variants are based on Chocolate, which does not provide any support for local communication. In the article, we compare Gianduja and its variants with a standard neuro-evolutionary approach. We consider a total of six different swarm robotics missions. We present results based on simulation and tests performed with 20 e-puck robots. Results show that, typically, Gianduja and its variants are able to associate a meaningful semantics to messages.

Introduction

Communication can have a major impact on the performance of groups of robots (Kirby, 2002). This is particularly true in swarm robotics, where robots are deemed to cooperate to effectively perform a mission that an individual would not be able to perform alone (Dorigo, Birattari & Brambilla, 2014). In general, designing collective behaviors for robot swarms is challenging (Brambilla et al., 2013). Including effective communication mechanisms in the design makes it even more challenging (Balch, 2004).

Automatic methods are a promising way for designing robot swarms (Francesca & Birattari, 2016). In automatic design, the design problem is reformulated as an optimization problem in which design choices are the parameters to be optimized. An optimization algorithm searches the space of the possible solutions to maximize a mission-specific performance metric. Most of the advances in the automatic design of robot swarms belong in the neuro-evolutionary approach; robots are controlled by an artificial neural network that is trained via an evolutionary algorithm—for example, see Jakobi, Husbands & Harvey (1995), Urzelai & Floreano (2000), Nolfi & Floreano (2000), Trianni (2008), Gauci et al. (2014) and Silva et al. (2015). Besides neuro-evolution, other approaches have been proposed. For example, Francesca et al. (2014a) used probabilistic finite-state machines as a control architecture for the individual robots, while Jones et al. (2016) and Kuckling et al. (2018) used behavior trees.

In this paper, we study the automatic design of collective behaviors for robot swarms that are capable of local communication. We consider robots that can locally exchange messages. Our goal is to define and study methods that design collective behaviors for these robots and automatically define a semantics for the messages they can exchange. We propose Gianduja, an automatic design method that extends Chocolate (Francesca et al., 2015) by adding messaging capabilities to the robots—Gianduja and Chocolate are described in the following. By extending the capabilities of Chocolate, we also intend to corroborate the hypothesis made in Francesca et al. (2014a) that adding bias to the design method, by restricting the representational power of the control architecture it optimizes, allows it to generate solution that are more robust to the so-called reality gap (Ligot & Birattari, 2019; Jakobi, Husbands & Harvey, 1995). In Gianduja, robots can exchange a single bit of information, which amounts to broadcasting a message or not. We consider also two variants of Gianduja: Gianduja2 and Gianduja3, in which robots can exchange two and three bits of information, respectively—details are provided in the following. We first study Gianduja on three missions that a robot swarm can perform relying on exchanging one bit of information. Then, we study Gianduja2 and Gianduja3 on more complex missions in which a one-bit communication is not sufficiently informative to allow the coordination of the swarm.

Using a neuro-evolutionary approach, Quinn (2001) has already shown that it is possible to assign an implicit semantics to a message via an automatic design process. Nonetheless, his result is only marginally relevant to our research as it concerns only a pair of robots rather than a swarm. Moreover, the result was obtained on a single specific mission and does not immediately generalize to other missions. The study was performed in simulation only. In this paper, we automatically design collective behaviors based on communication for six different missions and we test them on a swarm of 20 e-puck robots. For all missions considered, the automatic design methods are able to assign an implicit semantics to the messages that robots exchange.

The research we present in this article belongs in the so-called automatic off-line design (Francesca & Birattari, 2016; Birattari et al., 2019). We consider a number of missions and for each of them, the control software is designed automatically, without any human intervention. The design phase is performed in simulation and the control software generated is directly ported to the robots, without any manual modification, to be assessed. Each mission is formally specified through a performance measure, an arrangement of the environment, and the definition of an area from which the robots are deployed. We follow the tenets of the research in automatic off-line design, as defined in Birattari et al. (2019): “(i) automatic off-line design methods should not be mission-specific and should be able to address a whole class of missions without undergoing any modification; (ii) once a mission is specified, human intervention is not provided for in any phase of the design process”. This implies that the designer cannot either conceive a specific method for a given mission at hand or fit an existing one to it by, for example, fine-tuning the parameters of the optimization algorithm. Specifically, the designer cannot run the design process multiple times and assess the outcome using simulated or real-robot experiments to gain insight on the mission at hand for then using their understanding to improve the simulation models or to modify/enrich the objective function in order to bootstrap the design process or prevent that undesirable behaviors emerge.

Related work

The emergence of signaling, syntax, and language in artificial agents has been widely studied from the perspectives of language sciences and engineering—for example, Steels (1998), Billard & Dautenhahn (1999), Cangelosi (2001), Loula et al. (2010)). In swarm robotics, two forms of communication are considered; indirect and direct communication. The first mainly refers to a form of communication displayed by social insects and known as stigmergy. This type of communication occurs by modification of the environment (Deneubourg et al., 1990; Garnier, Gautrais & Theraulaz, 2007). Direct communication, on the other hand, implies direct exchange of information between agents. In robotics, this kind of communication is usually implemented using infrared messaging, radio signals, sound signals or visual color signals (Gutiérrez et al., 2009; Quinn, 2001; Trianni, Labella & Dorigo, 2004; Ducatelle et al., 2011; Ferrante et al., 2014; Campo et al., 2010). In this paper, we focus on the design of direct communication between robots of a swarm. In the following, we refer to direct communication as simply communication.

The literature is extensive when it comes to multi-robot systems that use a form of communication. However interesting, many of these studies are outside the scope of this paper because they concern systems in which communication is a priori designed—for example, Balch & Arkin (1994), Cao et al. (1997); Jones & Mataric (2004); Fong & Nourbakhsh (2009); Balch (2004), Werfel, Petersen & Nagpal (2014), Ducatelle et al. (2011); Ferrante et al. (2014); Campo et al. (2010). We focus here on studies that involve the automatic design of communication in swarm robotics. Most existing studies on communication in swarm robotics were conducted in the framework of neuro-evolutionary robotics (Nolfi & Floreano, 2000; Trianni, Labella & Dorigo, 2004; Wischmann & Pasemann, 2006; Trianni, 2008; Marocco & Nolfi, 2006a; Wischmann, Floreano & Keller, 2012; Trianni, 2014; Uno et al., 2011). Quinn and coworkers (2001, 2003) studied for the first time the emergence of communication between agents. In their studies, robots developed communicative behaviors by detecting certain motion patterns of their peers, without the use of a dedicated communication device. The robots established a social interaction and assumed leader and follower roles. In Nolfi (2005) and Marocco & Nolfi (2006b), a group of robots evolved to solve a collective navigation problem. Robots were controlled by a neural network and could use four different signals. Without directly rewarding communicative behaviors, the evolutionary process produced behaviors that make an effective use of communication. The system was tested in simulation and on a swarm of four robots. Floreano et al. (2007) also studied the evolution of communicative behaviors using neuro-evolutionary robotics. In their study, the robots used visual signals to communicate the location of a food source. In their experiments conducted with physical robots, communication increased the performance of the swarm with respect to a non-communicative swarm. In Ampatzis et al. (2008), two robots had to recognize features of their environment and react accordingly. The robots were controlled by neural networks and could use their on-board speakers and microphones to communicate using acoustic signals. During the evolutionary process, communication emerged as a result of the performance improvement it allowed, even though it was not directly rewarded. The authors tested the performance of the robots both in simulation and reality with two s-bot robots (Mondada et al., 2003). Trianni & Nolfi (2009) studied the emergence of syncing behaviors in robot swarms. In this work s-bots robots capable of sending a sound message had to synchronise their motion. The robot were controlled by a neural network trained with an evolutionary algorithm. The authors tested the scalability of the system in simulation with up to 96 robots and tested with two and then three real robots. The origin of communication from a neuro-evolutionary perspective was studied by Tuci (2009). The same two robots considered in the previous study had to perform a categorization task using communication based on acoustic signals. The evolutionary process produced behaviors that use communication to improve performance although communication was not explicitly rewarded. Experiments were conducted in simulation only. In Marocco & Nolfi (2006a), the authors studied the emergence of communication behavior in a swarm of four simulated robots tackling an aggregation task. The robots were controlled by a neural-network and had to equally divide themselves in the two circular spots placed in the arena. The analysis of the evolutionary process showed that the robots acquired the ability to solve the problem by developing an effective communication system. The authors point out that co-adaptation of communication and motor skills on the robot play a big role in the emergence of communication. In Uno et al. (2011) the authors extended the work of Marocco & Nolfi (2006a) by investigating the relationship between robots and conditions of emergence of communication from the point of view of language sciences. The setup and task in this study is similar to the one of Marocco & Nolfi (2006a). Simulation were conducted using a swarm of two robots conducting an aggregation task with color spot cues. The work of De Greeff & Nolfi (2010) also extends the one of Marocco & Nolfi (2006b). In their works the authors studied the importance of direct and indirect communication (namely explicit and implicit communication) by using bluetooth for direct communication and proximity sensors for indirect communication. The robots were controlled by a neural-network that was evolved using an evolutionary algorithm. Experiments with real robots were conducted with two e-puck robots. Communication was also studied by Wischmann, Floreano & Keller (2012), who compared different communication strategies evolved from the same initial population and highlighted the trade-off between communication efficiency and robustness. Experiments were run in simulation only.

The research we present in this paper is related to the one of Ampatzis et al. (2008) and Tuci (2009). Indeed, our goal is to define an automatic design process in which messages that do not have an a priori semantics are effectively used to communicate. Messages are associated with a semantics by the design process, on a per-mission basis.

GIANDUJA

Gianduja and all its variants that we will present in the following are based on Chocolate (Francesca et al., 2015). As Chocolate, they belong to the AutoMoDe class of methods originally defined by Francesca et al. (2014b). These methods automatically generate control software by assembling predefined, mission-agnostic software modules. Like Chocolate, Gianduja produces control software for the e-puck platform (Mondada et al., 2009), extended with three hardware modules: the Overo Gumstix, the ground sensor, and the range and bearing. The e-puck is a circular two-wheeled robot, whose diameter is about 70 mm. It has 8 IR transceivers, positioned all around its body, that double as light and proximity sensors. The Overo Gumstix module is a single-board computer that allows the e-puck to run Linux. The ground sensor module allows the e-puck to perceive the color of the floor. The range-and-bearing module (Gutiérrez et al., 2009) is an infrared communication device for local sensing and messaging. It operates by broadcasting every 100 ms a 2-byte ping: one byte encodes the sender ID and the other an arbitrary payload. The ping can be received by robots within a range of about 0.7 m from the sender. A robot that receives a ping is able to estimate the relative position of the sender in polar coordinates—range and bearing, hence the name of the device.

As in Chocolate, the control software also in Gianduja, has the form of a probabilistic finite-state machine, which is assembled using pre-existing, mission-independent modules. Modules are either behaviors to be used as states of the state machine or conditions to be associated with its edges. Conditions determine whether a transition should happen or not. Modules may have tunable parameters that modify their functioning. The topology of the probabilistic finite-state machine, the behaviors and the conditions to be included, and the value of their parameters are determined by an optimization algorithm that maximize a mission-specific performance measure. The optimization algorithm adopted in Gianduja, as well as in Chocolate, is iterated F-race (Balaprakash, Birattari & Stützle, 2007; Birattari et al., 2010). Within the optimization process, simulations are performed using ARGoS3 (Pinciroli et al., 2012).

The only element in which Gianduja differs from Chocolate is that in Gianduja the modules have parameters that control local communication between robots. Each robot can locally broadcast messages and receive those broadcast by neighboring peers. Messages are a priori pure signifiers: they are not associated to any semantics. It is the optimization algorithm that, on a per-mission basis, gives a meaning to messages, that is, associates a semantics to signifiers. In the context of this research, giving a meaning to a message means to define (i) the conditions under which the sender broadcasts the message and (ii) the effects that the message has on the recipient’s behavior.

In the following, we present the results of two studies. In Study A, we focus on the automatic design of collective behaviors for robots that locally broadcast one bit of information using the range-and-bearing module. In practice, robots communicate by setting a one-bit flag of the ping’s payload. A robot that perceives a neighboring peer via the range-and-bearing module—that is, that receives its ping—will read the flag: if it is set, the neighboring peer is broadcasting the message; otherwise, it is not.

In Study B, we generalize Study A to robots that locally broadcast two (or three) bits of information via the range-and-bearing module. In this case, a robot communicates by writing two (or three) bits of the ping’s payload. As a result, robots broadcast/react to three (or seven) distinct messages—that is, 22 − 1 (or 23 − 1), where the −1 accounts for the case in which a robot does not broadcast any message.

Experiments in Study A are performed with real robots and in the so-called pseudo-reality. A pseudo-reality is a simulation model, different from the one on which the design process has been performed. It has been shown that experimental runs in pseudo-reality provide indications on the intrinsic ability of a method to cross the reality gap successfully (Ligot & Birattari, 2019). After having validated the ability of pseudo-reality to provide reliable evaluations, we will adopt it as the exclusive device to perform the experiments on Study B. Performing these experiments with real robots would be time consuming and would not give any further contribution to the scientific message of the paper.

Study A: One bit of information.

In this section, we study the automatic design of behaviors for robots that exchange one bit of information.

Design methods

In Study A, we consider four design methods.

Chocolate

Chocolate (Francesca et al., 2015) generates probabilistic finite-state machines by assembling parametric modules—behaviors and conditions—that exploit the capabilities of the e-puck platform, as formally defined by the reference model RM1.1; further details are available in Hasselmann et al. (2018). It is important to notice that Chocolate’s modules are strictly part of the definition of Chocolate itself. They were defined once and for all in a mission-agnostic way (Francesca et al., 2014b). They are not supposed to be modified or integrated by other modules when Chocolate is applied to a specific mission. Yet, these modules have parameters that impact their functioning and that are automatically fine-tuned on a per-mission basis. In Chocolate, low-level behaviors are:

exploration: the robot performs a random walk, while avoiding obstacles;

stop: the robot stands still;

phototaxis: the robot goes towards the light source, if perceived;

anti-phototaxis: the robot goes in the opposite direction;

attraction-to-neighbors: the robot goes towards its neighboring peers;

repulsion-from-neighbors: the robot goes in the opposite direction.

Conditions are:

black-floor: change state if floor is black;

white-floor: change if it is white;

gray-floor: change if it is gray;

neighbor-count: change if sufficiently many neighboring peers are perceived;

inverted-neighbor-count: change if they are sufficiently few;

fixed-probability: change state with a fixed probability.

For details on these modules and their tunable parameters, we refer the reader to the original publication in which the modules were introduced (Francesca et al., 2014b).

The topology of the probabilistic finite-state machine, the modules to be included and their parameters are defined by an optimization process. The space of the probabilistic finite-state machines that Chocolate can possibly generate is constrained to those comprising at most four states having each at most four outgoing edges. As an optimization algorithm, Chocolate uses the implementation of iterated F-race provided by the R package irace (López-Ibáñez et al., 2016), with its default parameters. Iterated F-race is based on F-race (Birattari et al., 2002), a racing procedure (Maron & Moore, 1997) originally proposed for the automatic configuration of stochastic optimization algorithms and metaheuristics (Hoos & Sttzle, 2004; Gendreau & Potvin, 2019). In F-race, a set of candidate solutions are randomly sampled and then sequentially evaluated, over a set of test cases, to eventually select the most suitable one. Along the sequential evaluation of candidate solutions, a Friedman test is repeatedly performed to identify candidate solutions that perform significantly worse than at least another one. These solutions are discarded so that the evaluation can focus on the best ones. The algorithm terminates when only one candidate solution remains or when a predefined budget of evaluations is depleted. Iterated F-race consists of multiple iterations of F-race. After the first iteration, each subsequent one operates on a set of candidate solutions that are sampled around those that the previous iteration selected as the best ones. The algorithm terminates when a predefined budget of evaluations is depleted. Within the optimization process, simulations are performed using the ARGoS3 simulator (Pinciroli et al., 2012), version beta 48, together with the argos3-epuck library (Garattoni et al., 2015). ARGoS3 is a modular multi-physics robot simulator specifically conceived to simulate robot swarms. Chocolate uses ARGoS3’s 2D dynamic physics engine to simulate the robots and the environment. The argos3-epuck library provides low-level implementations of the sensors and actuators of the e-puck robot with fine control on noise levels for all actuators and sensors. ARGoS3 and the argos3-epuck library inject a realistic level of sensor and actuator noise in all simulations as suggested by Miglino, Lund & Nolfi (1995) as a good practice for reducing the impact of the reality gap.

As Chocolate is unable to produce behaviors that leverage communication, it will likely fail to be effective on the missions we consider in the following of the paper. We include it in the analysis to act as a baseline, so as to appraise the importance of communication.

EvoCom

EvoCom is an automatic design method that extends EvoStick (Francesca et al., 2014a) by adding communication capabilities. EvoStick is a standard neuro-evolutionary robotics method. It was used in Francesca et al. (2014a) as a yardstick against which the authors compared their newly proposed method, but had been previously analyzed in Francesca et al. (2012). EvoStick was then included in other empirical studies (Francesca et al., 2014a; Francesca et al., 2015; Birattari et al., 2010; Ligot & Birattari, 2019). We chose EvoStick as a starting point for our development because, to the best of our knowledge, it is the only neuro-evolutionary method for the design of robot swarms that has so far been studied following the tenets of off-line automatic design (Birattari et al., 2019), as defined in the Introduction. Namely, it is the only neuro-evolutionary design method that has been tested on multiple missions without any mission-specific modification so as to evaluate its ability to generate control software without any human intervention. The goal for EvoCom is to introduce a comparison point with Gianduja, the idea is not to find the best possible neural network topology for a given mission but rather to use a general purpose topology that could work with any given mission.

EvoCom creates a feed-forward fully-connected neural network with no hidden nodes, which comprises 30 input nodes and 3 output nodes. Inputs and outputs are based on the elements of the reference model RM2, see Table 1.

Table 1 Reference model RM 2.

Novelties are highlighted with respect to RM 1.1. They concern the ability to send and react to one bit of information. Vb is computed in the same way as V by restricting to broadcasting neighboring robots.

Input	Value	Description	
proxi∈{1,…,8}	[0,1]	reading of proximity sensor i	
lighti∈{1,…,8}	[0,1]	reading of light sensor i	
gndj∈{1,2,3}	{black, gray, white}	reading of ground sensor j	
n	[0,19]	number of neighboring robots perceived	
V	([0.5, 20], [0, 2π])	their relative aggregate position	
b	[0,19]	number of messaging neighbors perceived	
Vb	([0.5, 20], [0, 2π])	their relative aggregate position	
Output	Value	Description	
vk∈{l,r}	[ − 0.12, 0.12]	target linear wheel velocity	
s	{on, off}	broadcast state	
Notes.

Period of the control cycle: 100 ms.

The inputs are: 8 proximity sensors, 8 light sensors, 3 ground sensors, 10 values computed based on data from the range-and-bearing module, and 1 bias. The outputs are: 2 for wheel speed and 1 for broadcasting the message. The proximity sensors, light sensors, and ground sensors directly feed their value, as defined in RM2, to the corresponding inputs of the neural network. The range-and-bearing module determines the values of 10 inputs: 5 inputs concern the detection of neighboring peers and are computed on the basis of n and V; the other 5 concern messages received and are computed on the basis of b and Vb. Also in the case of n, V, b, and Vb, the definition is given in RM2. One of the five inputs devoted to neighboring peers equals zn=1−21+en; the other four are the scalar projection of V onto four unit vectors that point at 45°, 135°, 225°, and 315° with respect to the front of the robot. One of the five inputs devoted to messages equals zb=1−21+eb; the other four are the scalar projection of Vb onto the aforementioned four unit vectors. The two output nodes encoding wheel speed range in [−0.12,0.12] m/s . The one devoted to the message takes a binary value: 1 if the message should be broadcast, 0 otherwise. The activation of the output neurons is computed as the weighted sum of all input units plus a bias term, filtered through a logistic function. All synaptic weights are real values in the range [ − 5, 5] and are optimized with an evolutionary algorithm. The initial population comprises 100 randomly generated individuals. At each iteration, each individual is evaluated 10 times in simulation. A new population is obtained via elitism and mutation: the best 20 individuals are kept unmodified, while the other 80 are obtained by mutation of the 20 best ones. The evolutionary algorithm stops when a predefined budget of evaluations is depleted. All simulations are performed using ARGoS3 and the argos3-epuck library under the same conditions and with the same noise levels as in Chocolate.

Gianduja

As already mentioned, Gianduja is the main method that we propose in this paper. It addresses the limitation of Chocolate regarding local communication: in Chocolate, robots cannot explicitly communicate and are only capable of detecting the presence of peers in their neighborhood. In Gianduja, as this method is based on the same reference model as EvoCom, namely RM2, robots are able to locally broadcast one message (one bit of information) and react to it. Messages are sent via the range-and-bearing module by setting a bit of the ping’s payload. Robots can thus identify their neighbors and estimate their position.

Gianduja generates control software for the e-puck platform, as formally specified by RM2, which is the same reference model adopted in EvoCom. Gianduja operates on eight low-level behaviors. Six are the same of Chocolate, extended with a binary parameter m: if m = 1, the message is broadcast while the behavior is performed; otherwise, it is not. The other two are:

attraction-to-message: the robot goes towards messaging peers;

repulsion-from-message: the robot goes in the opposite direction.

The direction of the messaging peers is provided by Vb, as defined in RM2. Behavior transitions can be triggered by eight conditions. Six are the same of Chocolate, the other two are:

message-count: change state if sufficiently many peers are messaging;

inverted-message-count: change if they are sufficiently few.

Under the message-count condition, a state transition happens with probability zb=11+eηξ−b; under inverted-message-count, the probability is z ¯b=1−zb. Here, η and ξ are parameters of the behavior to be assigned by the design process.

In all other respects, Gianduja is identical to Chocolate. In particular, the two methods adopt the same optimization algorithm with the same parameters, the same simulator with the same noise levels, and the same constraints on how probabilistic finite-state machines are assembled.

GiandujaE

GiandujaE is derived from Gianduja and differs from it only in the way received messages are handled. GiandujaE does not use conditions message-count and inverted-message-count. The mechanism for handling the number of messages received is embedded in all remaining conditions, which are extended with two parameters: message ID ν and threshold τ. Message ID can take two values: if ν = 0, the condition behaves as in Gianduja; if ν = 1, the condition is enabled and behaves as in Gianduja only if the number of neighboring peers that are messaging is larger than the threshold τ, which can take an integer value between 0 and 20.

In the following of the paper, a letter E in the name of the method indicates that received messages are handled as described above—the letter E is the mnemonic for embedded. In principle, this way of handling messages allows more complex behaviors to be obtained. It should be expected that the conjunction of a condition along with the presence of a message allows probabilistic finite-state machine of simpler structure, without sacrificing capabilities.

Experimental setting

We test and compare the four design methods described above on three different missions.

Missions

In all missions, robots operate in a dodecagonal arena of 4.91 m2. The arena is surrounded by walls. The floor is gray, although some areas may be black or white, depending on the specific mission. Details are provided in the following. All mission are intended to be performed by a swarm of N = 20 robots within T = 120 s.

The three missions are AGGREGATION, STOP, and DECISION. We select them because, a priori, one could imagine that communication would play a different role in each of them. Indeed, it is reasonable to expect that AGGREGATION can be solved without the use of communication but that communication can contribute to increasing performance. On the other hand, it is reasonable to expect that DECISION and STOP strictly require communication to be performed. It is also reasonable to expect that the semantics that the optimization would associate to the message is different for each mission.

AGGREGATION. Two circular spots of diameter 0.6 m mark the floor of the arena: one is white and the other black. They are positioned on the left-hand half of the arena, 0.25 m apart. At the beginning of each run, the robots are randomly positioned in the right-hand half of the arena so that no robot is on the spots—see Figs. 1A, 1D. The robots must aggregate on the white spot as quickly as possible. The black spot is included in the arena to acts as a possible disturbance to the automatic design process.

Figure 1 Arenas for the three missions: AGGREGATION (A,D), STOP(B,E), and DECISION (C,F); simulation (A,B,C) and real setup (D,E,F).

In DECISION, the central spot may be either black or whitewith equal probability.

Performance is measured by the following objective function—the higher, the better: Ca=24000−∑t=1T ∑i=1NIit;Iit=0,if robotiis on the white spot;1,otherwise.

Performance is non-negative and its theoretical maximum is 24000.

STOP. One circular white spot with diameter 0.2 m is located near the walls of the arena, in the top-left quadrant. The spot is smaller than in the other missions so that it is more challenging for the robots to find it. At the beginning of each run, the robots are randomly positioned in the right-hand half of the arena so that none of them is on the spot—see Figs. 1B, 1E). The robots must find the spot as quickly as possible and stop right after. Performance is measured by the following objective function—the higher the better: Cs=48000−t ¯N+ ∑t=1t ¯ ∑i=1NI ¯it+ ∑t=t ¯+1T ∑i=1NIit;

Iit=1,if robotiis moving;0,otherwise;I ¯it=1−Iit.

Here, t ¯ is the time at which the first robot finds the white spot. In the definition of Ii (and I ¯i), a robot is moving if it traveled more than 5 mm in the last 100 ms. We adopted this definition to avoid penalizing EvoCom which is unable to generate behaviors in which robots can stop still, due to the way the outputs of the neural network are encoded. Performance is non-negative and its theoretical maximum is 48000.

DECISION. One circular spot with diameter 0.6 m is located in the center of the arena. In each experimental run, the spot can be either black or white, with equal probability. A light source is positioned outside the arena, at its right. At the beginning of each run, the robots are randomly positioned in the arena—see Figs. 1C, 1F. The robots must gather in the right-hand half of the arena if the spot is white, or in the left-hand side if it is black. The performance measure—the higher, the better—is: CD=24000−∑t=1T ∑i=1NIit;

Iit=0,if robotiis in the correct half of the arena;1,otherwise.

Performance is non-negative and its theoretical maximum is 24000.

Protocol

We compare four design methods on three missions. All experiments involve a swarm of 20 e-puck robots. For each mission, each method is executed 14 times so as to obtain 14 instances of control software. Each execution is allowed a budget of 200000 simulation runs. We evaluate each method by running the 14 instances of control software it produced once in simulation and once in pseudo-reality—explanation follows1. For Chocolate, Gianduja, and EvoCom, the control software produced is run once also on real robots. The order of the experimental runs is randomized to avoid any bias. The performance of the real robots is automatically computed by a tracking system based on an overhead camera (Stranieri et al., 2013). The simulation model used as a pseudo-reality for the evaluations of the control software produced is available as Supplemental Information (Hasselmann & Birattari, 2019).

Statistics.

For each mission, we use boxplots for reporting the performance recorded in simulation using the same model adopted in the design process, in pseudo-reality, and on real robots—when relevant. Simulation results (either on the design model or in pseudo-reality) are represented by gray boxes: thin boxes for the results obtained on the design model and thick ones for those obtained in pseudo-reality. Real-robot results are represented by thick white boxes. Statements on the relative performance of two methods, are supported by a Wilcoxon rank-sum test, at 95% confidence (Conover, 1999): any statement like “A performs significantly better/worse than B” implies that a Wilcoxon rank-sum test was employed and detected significance with confidence of at least 95%.

We eventually perform a Friedman test (Conover, 1999), which aggregates all results by ranking the performance of all methods across all missions. We report the results in a plot that represents the average rank of each method and its 95% confidence interval. The performance of two methods is significantly different if the corresponding intervals do not overlap. In the context of an aggregate analysis, any statements like “A performs significantly better/worse than B” will be based on the observation that the corresponding intervals do not overlap and will be therefore valid with confidence of at least 95%.

In the whole paper, the adjective significant and the adverb significantly are used only to refer to statistical significance. Whenever we use these terms, we imply that an appropriate statistical test (either Wilcoxon or Friedman) has been performed.

Results

Numerical results, videos, code, and probabilistic finite-state machines generated by Chocolate and Gianduja {ø,E} are available as Supplemental Information (Hasselmann & Birattari, 2019). In the following, with the notation Gianduja {ø,E}, we will collectively refer to Gianduja, and GiandujaE.

AGGREGATION. Results are reported in Fig. 2A. Both Gianduja and Chocolate perform significantly better than EvoCom. Although Gianduja is significantly better than Chocolate on the simulation model used for the design, the two perform similarly in reality. In this particular mission, there is no evidence that GiandujaE improves over Gianduja. EvoCom suffers more from the (pseudo-) reality gap than the other methods. Gianduja suffers more from the reality gap than Chocolate: this could be due to the uncertainty of Vb. Although the ability to detect where a message originates is a potential advantage, it becomes a burden due to uncertainty.

Figure 2 AGGREGATION (A), STOP (B), and DECISION (C).

Thick white boxes represent the results of robot experiments; thick gray boxes those of pseudo-reality; thin gray ones, those of simulations performed on the basis of the same simulation model adopted in the design process. The higher the better.

At visual inspection, in Chocolate robots navigate randomly in the arena and stop whenever they enter the white spot. In Gianduja {ø,E}, when this happens, robots start broadcasting the message to attract neighboring robots. In EvoCom robots randomly explore the arena until they enter the white spot. When this happens, they spin in place. Videos are available in Hasselmann & Birattari (2019).

In this mission, pseudo-reality gives a reasonably good indication on the performance in reality of EvoCom and Chocolate but overestimates the performance of Gianduja.

An example of probabilistic finite-state machine produced by Gianduja is reported as Supplemental Information (Hasselmann & Birattari, 2019). The robots start in the exploration state. If they step on white ground, they can transition to stop and start broadcasting the message. Robots that are still in the exploration state and receive a sufficiently large number of messages to trigger the message-count condition transition to the attraction to message behavior. As a result, they will approach the robots that broadcast the message, eventually reaching the white spot. Seen from the point of view of the sender, the semantics given to the message is: I am on the white spot; while, seen from the point of view of the receiving robots, it is: Go there.

STOP. Results are reported in Fig. 2B. Gianduja performs significantly better than EvoCom and Chocolate on the simulation model used for the design and in reality. In this mission, Gianduja performs significantly better than GiandujaE in pseudo-reality.

At visual inspection, EvoCom appears to be unable to use communication effectively whereas Gianduja {ø,E} do. In EvoCom robots move randomly until their motion is stopped by the walls of the arena. This behavior is qualitatively similar to the one of Chocolate, which cannot leverage communication. In Gianduja {ø,E}, robots move randomly until one of them enters the white spot, when this happens the robot stops and starts sending the message. Other robots then stop and relay the message. Videos are available in Hasselmann & Birattari (2019).

Chocolate and EvoCom suffer the reality gap more than Gianduja. The performance in pseudo-reality gives a good indication on the relative drop in performance of the three methods but, in this case, fails to predict the high variance that we observe in reality.

An example of probabilistic finite-state machine produced by Gianduja is reported as Supplemental Information (Hasselmann & Birattari, 2019): the robots start by performing exploration; they transition to stop either if they step on white floor or if they receive the message from their peers; while in stop, they broadcast the message. Seen from the point of view of the sender, the semantics given to the message is: I found the white spot; while, seen from the point of view of the receiving robots, it is: Stop.

DECISION. Results are reported in Fig. 2C. Gianduja performs significantly better than EvoCom and Chocolate on the simulation model used for the design, in pseudo-reality, and in reality. In pseudo-reality, GiandujaE is slightly but significantly better than Gianduja. Also, the inter-quartile range observed in this experiment is smaller for GiandujaE than for the other two methods. Both GiandujaE and Gianduja are significantly better than Chocolate in pseudo-reality. This indicates that, in this mission, communication contributes to performances and Gianduja {ø,E} leverage it effectively.

At visual inspection, it appears that Gianduja {ø,E} use communication meaningfully. In all these methods, every realization of the design process selects a default behavior: by default, at the beginning of a run, the robots either go towards the light or away from it. While doing so, some robots by chance enter the central spot: if the color does not correspond to the default behavior, they start broadcasting the message and revert their direction of motion. Robots that receive the message relay it and revert their direction of motion as well. In Chocolate, to make an informed decision, each individual robot needs to enter the central spot. In EvoCom, robots randomly select one side or the other of the arena. As a result, performance is very inconsistent. Videos are available in Hasselmann & Birattari (2019).

Gianduja {ø,E} cross the (pseudo-) reality gap in a satisfactory way. In this mission, pseudo-reality gives a good indication of the performance in reality.

An example of probabilistic finite-state machine produced by Gianduja is reported as Supplemental Information (Hasselmann & Birattari, 2019): the robots start in the anti-phototaxis behavior; they transition to phototaxis either if they step on white floor or if they receive the message from their peers; while in phototaxis, they broadcast the message. Seen from the point of view of the sender, the semantics given to the message is: I stepped on the white spot; while, seen from the point of view of the receiving robots, it is: Go towards the light.

Aggregate results

The aggregate results obtained in real-robot experiments are presented in Fig. 3; those obtained in pseudo-reality are presented in Fig. 4. Across all missions, the Friedman test indicates that, on the robots, the control software generated by Gianduja performs significantly better than the one of Chocolate and EvoCom. In pseudo-reality, Gianduja performs significantly better than EvoCom and Chocolate. All in all, on the aggregate data, pseudo-reality provides a correct prediction of the relative performance in robot experiments.

Figure 3 Friedman test on the aggregate results of the three missions.

The plot represents the average ranks of the three methods and their 95% confidence interval in robot experiments; the lower the better.

Figure 4 Friedman test on the aggregate results of the three missions.

The plot represents the average ranks of the four methods and their 95% confidence interval in pseudo-reality experiments; the lower the better.

The aggregate results over the three missions confirm that communication improves performance and that Gianduja leverages it effectively yielding significantly better results than EvoCom. At least in the setting considered in our experiments, EvoCom was unable to evolve effective behaviors. The performance of the behavior produced by EvoCom is worse than the one produced by Chocolate, which by construction does not use communication. This can be considered as a major failure. The results on the three missions considered fail to provide any evidence that GiandujaE is better than Gianduja.

Study B: Communication with two or three bits of information.

In this section, we study the automatic design of communication when robots can broadcast and react to multiple messages. We introduce a number of automatic design methods that extend those we described in Study A by enabling the possibility of exchanging more information. Specifically, we introduce Gianduja {2,2E} and Gianduja {3,3E}, which extend Gianduja {ø,E} with the ability to broadcast and react to two or three bits of information, respectively. Gianduja {2,2E} are based on reference model RM2.1-2; Gianduja {3,3E}, on RM2.1-3—see Table 2.

Table 2 Reference model RM 2.1-ℓ.

Novelties with respect to RM 2 are highlighted. RM 2.1-ℓ is a parametric family of models that differ one from the other in the amount of information that robots can exchange. The parameter ℓ identifies a specific model within the family by defining the number of bits of information that robots can exchange: in RM 2.1-2, robots can exchange two bits of information; in RM 2.1-3, three. RM 2.1-ℓ is a proper extension of RM 2: for ℓ = 1, RM 2.1-1≡ RM 2.

Input	Value	Description	
proxi∈{1,…,8}	[0,1]	reading of proximity sensor i	
lighti∈{1,…,8}	[0,1]	reading of light sensor i	
gndj∈{1,2,3}	{black, gray, white}	reading of ground sensor j	
n	[0,19]	number of neighboring robots perceived	
V	([0.5, 20], [0, 2π])	their relative aggregate position	
bh∈{1,…,2ℓ−1}	[0,19]	number of neighbors broadcasting message h	
Vh∈{1,…,2ℓ−1}	([0.5, 20], [0, 2π])	their relative aggregate position	
Output	Value	Description	
vk∈{l,r}	[ − 0.12, 0.12]	target linear wheel velocity	
sh∈{1,…,2ℓ−1}	{on, off}	broadcast state for message h	
Notes.

Period of the control cycle: 100 ms.

We evaluate these methods on new missions that we think require the use of at least two different messages. The goal of the experiments is to assess whether Gianduja, and more generally the ideas presented in the paper, can possibly scale up to more complex missions that require more information to be exchanged by the robots. We also wish to see if the way in which messages are handled by GiandujaE increases the performance with respect to Gianduja when missions are more complex than those considered in Study A. Finally, we wish to understand whether the design methods proposed in the paper are able to handle the larger search space that results from increasing the number of messages that robots can broadcast and to which they can react.

Design methods

In this section, we introduce six new design methods based on Gianduja, GiandujaE, and EvoCom.

Gianduja2

Gianduja2 extends Gianduja by allowing the communication of two bits of information, which amounts to the ability to broadcast and react to 22 − 1 = 3 different messages. The term −1 accounts for the possibility of not sending any message. Gianduja2 operates on the same low-level behaviors of Gianduja, with only a minor modification. Here, the parameter m can take four integer values: {0, …, 3}. If m = 0, the robot does not broadcast any message—or, more precisely, it broadcasts via its range-and-bearing module the null message that only allows its neighboring peers to detect its presence, without conveying any extra signification. If m = {1, 2, 3}, the robot broadcasts the corresponding message, to convey the meaning that the optimization process will have automatically associated with the message itself, for the specific mission at hand. Moreover, in Gianduja2, the two behaviors attraction-to-message and repulsion-from-message take one additional parameter m¯∈1,2,3, which specifies the message to whose senders the robot is attracted, or from whose senders it is repelled. Finally, the two conditions message-count and inverted-message-count take one additional parameter m¯∈1,2,3, which specifies the message to which the condition should be sensitive. The transition probability is computed via z(⋅), as defined in Gianduja.

Gianduja2E

Gianduja 2E derives from Gianduja2, in the same sense in which GiandujaE derives from Gianduja. Like GiandujaE, Gianduja 2E does not use the conditions message-count, and inverted-message-count. Also in Gianduja 2E, the conditions black-floor, white-floor, gray-floor, neighbor-count, inverted-neighbor-count, and fixed-probability are extended with two parameters: message ID ν and threshold τ. Parameter ν can take four different values: if ν = 0, the condition behaves as in Gianduja2; if ν = {1, 2, 3}, the condition is enabled and behaves as in Gianduja2 if the number of neighbors sending message 1, 2, 3, respectively, is greater than τ.

EvoCom2

EvoCom2 extends EvoCom by allowing the communication of two bits of information. With respect to EvoCom, EvoCom2 has five additional inputs and one additional output: the network totalizes 35 inputs and 4 output nodes. The five additional inputs are used to encode information concerning the messages received. In EvoCom, five inputs are devoted to this; in EvoCom2, they are ten. Two are computed on the basis of the number of neighboring peers that broadcast the various messages available, and the remaining eight on the basis of the direction from which messages arrive. The additional output is used to encode the broadcast state. In EvoCom, one output is devoted to this; in EvoCom2, they are two.

Before we can formally describe how these inputs and outputs are computed, we need to define some notation. In EvoCom2, messages are identified via a binary representation. Table 3 provides a mapping between a message ID h and its binary representation. Let us define the function bitp(h) that returns TRUE/1, if the p-th bit of the binary representation of h is set; and FALSE/0, otherwise. In EvoCom2, p ∈ {0, …, ℓ − 1}, with ℓ = 2. Bit p = 0 is the less significant one. For p ∈ {0, …, ℓ − 1}: let γp = ∑h:phb(h) be the number of neighboring peers broadcasting any message h whose binary representation has the p-th bit set; and let Γp = ∑h:phV(h), be the composition of vectors Vh concerning any message h whose binary representation has the p-th bit set.

With this notation, we can represent the ten inputs devoted to encoding information concerning the messages received: two are zγp=1−21+eγp, with p ∈ {0, …, ℓ − 1}; eight are the projections of Γp, with p ∈ {0, …, ℓ − 1}, onto the unit vectors pointing at 45°, 135°, 225°, and 315° with respect to the front of the robot. The two outputs determine which message should be broadcast according to the binary representation given in Table 3.

Gianduja3

Gianduja3 extends Gianduja2, and eventually Gianduja, by allowing the communication of three bits of information, which amounts to the ability to broadcast and react to 23 − 1 = 7 different messages. The term −1 accounts for the possibility of not sending any message. Gianduja3 is defined as Gianduja2, with the difference that the parameter m can take eight integer values, {0, …, 7}; and the parameter m¯ can take seven: {1, …, 7}.

Table 3 Binary encoding of messages adopted in EvoCom2—white background: 2 bits for the 4 messages, including the null one.

Encoding adopted in EvoCom3—white and gray background: 3 bits for the 8 messages, including the null one.

h	bit2(h)	bit1(h)	bit0(h)	
0	0	0	0	
1	0	0	1	
2	0	1	0	
3	0	1	1	
4	1	0	0	
5	1	0	1	
6	1	1	0	
7	1	1	1	

Gianduja3E

Gianduja 3E derives from Gianduja3, in the same sense in which GiandujaE and Gianduja 2E derive from Gianduja and Gianduja2, respectively. Gianduja 3E is defined as Gianduja 2E, with the difference that the parameter ν can take eight integer values, {0, …, 7}.

EvoCom3

EvoCom3 extends EvoCom2, and eventually EvoCom, by allowing the communication of three bits of information. It is defined as EvoCom2, with ℓ = 3. Here, the total number of inputs is 40 and the total number of output is 5. The number of inputs used to encode the number of neighboring peers that broadcast the various messages available is ℓ = 3. The number of inputs used to encode the direction from which messages arrive is 4ℓ = 12. The number of outputs used to encode the broadcast state is ℓ = 3.

Experimental setting

We test and compare Gianduja {2,2E,3,3E} and EvoCom {2,3} on 3 different missions.

Missions

In all three missions, N = 20 robots operate in the same dodecagonal arena described in Study A. As in Study A, the amount of time available to the robots for performing each mission is T = 120 second. The three missions are AGGREGATION, STOP, and DECISION. They were selected because we thought that, to be successfully performed, they require the use of multiple messages with different semantics. In all three missions, there is a beacon that can broadcast a message to which robots must react. The role of the beacon is played by an e-puck robot that does not move throughout the experiment. It is not considered to be part of the swarm.

BEACON AGGREGATION. Two circular spots of diameter 0.6 m mark the floor of the arena: one is white and the other black. They are positioned on the left-hand half of the arena, separated by 0.25 m—see Fig. 5A. A beacon positioned between the two spots broadcasts either message 1 or 2 for the whole duration of a run. The message to be broadcast is selected randomly with equal probability at the beginning of each run. At the beginning of each run, the robots are randomly positioned in the right-hand half of the arena so that no robot is on either spot. Robots must aggregate on the black spot, if the beacon broadcasts 1; on the white one, if it broadcasts 2.

Figure 5 Arenas for the three missions: BEACON AGGREGATION (A), BEACON STOP (B), and BEACON DECISION (C); simulation setup.

The beacon is circled in red.

Performance is measured by the following objective function—the higher the better: CBA=24000−∑t=1T ∑i=1NIit;

Iit=0,if robotiis on the correct spot;1,otherwise.

Performance is non-negative and its theoretical maximum is 24000.

BEACON STOP. The arena’s floor is gray and a beacon is positioned in the middle of the arena—see Fig. 5B. The beacon broadcasts message 1 or 2 after a time uniformly sampled in [40 s; 60 s]—the message to be broadcast is selected randomly with equal probability at every run. At the beginning of each run, the robots are randomly positioned in the arena. Robots must stop as soon as the beacon starts broadcasting either one of the two messages.

Performance is measured by the following objective function—the higher the better: CBS=24000−∑t=1t ¯ ∑i=1NI ¯it+ ∑t=t ¯+1T ∑i=1NIit;

Iit=1,if robotiis moving;0,otherwise;I ¯it=1−Iit.

In this equation, t ¯ is the time at which the beacon starts broadcasting a message. In the definition of Ii (and I ¯i), a robot is considered moving if it traveled more than 5 mm in the last 100 ms. This accounts for the inability of EvoCom to generate behaviors in which robots can stop still. Performance is non-negative and its theoretical maximum is 24000.

BEACON DECISION. The floor of the right-hand quarter of the arena is black; the rest is gray. A light source is positioned outside the arena on its right-hand side—see Fig. 5C. A beacon is positioned at the middle of the interface between the black and the gray parts and acts as a timed trigger. At the beginning of a run, the beacon starts broadcasting either message 1 or 2—randomly selected with equal probability. After 60 S, the beacon switches to the other message. At the beginning of each run, the robots are randomly positioned in the arena.

Robots must position themselves either on the black or on the gray part of the arena, depending on the message broadcast by the beacon: on the black, if the message is 1; on the gray, if it is 2. The performance of the robots is measured by the following objective function—the higher, the better: CBD=24000−∑t=1T ∑i=1NIit;

Iit=0,if  robotiis  in  the  correct  side  of  the  arena;1,otherwise.

Performance is non-negative and its theoretical maximum is 24000.

Protocol

The protocol is similar to the one of Study A. We compare six design methods on three missions. All experiments involve a swarm of 20 e-puck robots. For each mission, each method is executed 14 times so as to obtain 14 instances of control software. Each execution is allowed a budget of 200000 simulation runs. We evaluate each method by running the 14 instances of control software it produced once in simulation and once in pseudo-reality (Ligot & Birattari, 2019).

Statistics.

Results are analyzed and reported as described in Study A.

Results

We present the results on a per-mission basis and then we aggregate them across the three missions. Numerical results, videos, code, and probabilistic finite-state machines generated by Gianduja {2,2E,3,3E} are available as Supplemental Information (Hasselmann & Birattari, 2019).

Results are reported in Fig. 6A. Gianduja and all its variants perform significantly better than EvoCom in pseudo-reality. All methods derived from Gianduja are also less affected by the pseudo-reality gap than EvoCom. At visual inspection, all methods derived from Gianduja perform similarly in simulation and pseudo-reality. Although the robots aggregate on the spots, they do not comply with the instructions provided by the beacon and split evenly between the two spots. Videos are available in Hasselmann & Birattari (2019). The poor performance on this missions may be due to the complexity of the task and the absence of any memory component in all the control architectures considered.

Figure 6 BEACON AGGREGATION (A), BEACON STOP(B), and BEACON DECISION(C).

Thick boxes represent the results in pseudo-reality; thin ones, those of simulations performed on the same simulation model adopted in the design process; the higher, the better.

An example of probabilistic finite-state machine produced by Gianduja 2E is reported as Supplemental Information (Hasselmann & Birattari, 2019): as the robots start on gray floor, they immediately transition from stop to exploration. They then transition back to stop whenever they find either the white or the black spot, no matter the message broadcast by the beacon. In this mission, the semantics assigned to the message by the optimization algorithm is unclear. The robots fail to use their messaging capabilities effectively. Our conjecture is that the probabilistic finite-state machines adopted by Gianduja {2,2E,3,3E} are not sufficiently expressive: the maximum number of states and transitions allowed in the design process is too small. We also conjecture that the duration of the experiment is too short for the robots to be able to accomplish the mission. We elaborate on these conjectures in the section further analysis of the Supplemental Information (Hasselmann & Birattari, 2019).

BEACON STOP. Results are reported in Fig. 6B. Gianduja {2E,3E} outperform all other methods both in simulation and pseudo-reality. Gianduja {2,3} perform significantly better than EvoCom in pseudo-reality. EvoCom suffer greatly from the pseudo-reality gap, Gianduja {2,3} are less affected, and Gianduja {2E,3E} are almost not affected at all. At visual inspection, all methods derived from Gianduja appear to correctly associate the messages broadcast by the beacon to stop. The structure of the state machines produced by Gianduja {2E,3E} are simpler, more robust, and produce better results than those produced by Gianduja {2,3}. In Gianduja {2,2E,3,3E}, the robots randomly explore the arena until they receive a message from the beacon. When this happens, they stop and start relaying the message. All in all, the robots use their communication capabilities effectively. Videos are available in Hasselmann & Birattari (2019). As in STOP of Study A, EvoCom {2,3} produce behaviors in which robots move randomly until their motion is stopped by the walls of the arena. Although trivial, this behavior performs reasonably well because, due to the size of the arena, the typical time needed by the robots to reach the walls is close to t ¯. Based on this observation, we conjecture that the performance of the behaviors generated by EvoCom might not be robust to variations in the arena size. We elaborate on this conjecture in the section further analysis of the Supplemental Information (Hasselmann & Birattari, 2019), where we compare the behaviors generated by EvoCom 2 and Gianduja 2E in arenas of different sizes.

For all methods considered, the performance of the three-bit versions is relatively close to the one of their two-bit counterparts. We can therefore conclude that, in this mission, the respective optimization algorithms can effectively search the larger solution space that results from extending the amount of information exchanged to three bits.

An example of probabilistic finite-state machine produced by Gianduja 2E is reported as Supplemental Information (Hasselmann & Birattari, 2019): the robots start in phototaxis—which, in the absence of a light source, reduces to exploration. They transition to stop whenever they receive message 1 from the beacon and the floor is gray (which is always true) or when they receive message 2 and sufficiently few robots are in the neighborhood (which is easily achieved considering the parameters of the behavior). While in the stop state, robots broadcast message 2 to trigger the transition on other robots. Seen from the point of view of the sender, the semantics given to the message is: I received a message (I am relaying it); while, seen from the point of view of the receiving robots, it is: Stop.

BEACON DECISION. Results are reported in Fig. 6C. At visual inspection, Gianduja {2E,3E}, and EvoCom appears to make an effective use of communication. Depending on the message broadcast by the beacon, the robots are either attracted or repulsed by the light and thus go towards or away from the black area. At time T=t ¯, when the beacon switches message, the robots invert their behavior. Videos are available in Hasselmann & Birattari (2019).

Contrary to Gianduja {2E,3E}, Gianduja {2,3} were to be unable to generate swarms that can switch their behavior and leave the black area when requested to do so by the beacon. At least in this mission, Gianduja {2,3} are unable to produce behaviors as rich and expressive as those produced by Gianduja {2E,3E}. EvoCom suffer greatly from the pseudo-reality gap. Also in this mission, the increase of the number of bits is not detrimental to the performance: the performance of Gianduja3 (Gianduja 3E) is close to the one of Gianduja2 (Gianduja 2E).

An example of probabilistic finite-state machine produced by Gianduja 2E is reported as Supplemental Information (Hasselmann & Birattari, 2019): the robots start in phototaxis. If the beacon broadcasts message 1, the robots directly transition to anti-phototaxis, relay message 1, and oscillate back and forth—they do so by cycling through phototaxis, anti-phototaxis, and exploration. Otherwise, if the beacon broadcasts message 2, the robots cycle through phototaxis and exploration, depending on the number of neighbors and the color of the floor. By doing so they leave the black area. In this case, the state machine produced by Gianduja 2E is hardly readable due to the high number of transitions. It is however possible to infer a semantics for messages 1 and 2: Seen from the point of view of the sender, the semantics is: I received message 1 (or 2); while, seen from the point of view of the receiving robots, it is: Go away from the light for message 1 and Go towards the light for message 2.

Aggregate results

Figure 7 reports the aggregate results in pseudo-reality. They confirm that Gianduja {2,2E,3,3E} perform significantly better than EvoCom. The results also show that the addition of a third bit of data is not needed to perform the three missions considered: indeed, Gianduja {2,2E} perform better than Gianduja {3,3E}. Nonetheless, it is interesting to observe that Gianduja could handle nicely the increased search space resulting from the addition of the third bit. Although the performance of Gianduja3 (Gianduja 3E) is lower than the one of Gianduja2 (Gianduja 2E), the drop is relatively small: in Table 4, we report the relative median performance of Gianduja {3,3E} with respect to Gianduja {2,2E} and we compare it with the one of EvoCom2.

Conclusions

In the paper, we studied the automatic design of control software for a swarm of robots capable of local communication. Specifically, we studied the emergence of a semantics for an a priori meaningless message: the automatic design process is expected to assign an appropriate semantics to a message—intended here as a pure signifier—based on the specific mission to be performed. We introduced Gianduja and its variants, a family of automatic design methods that produce control software in the form of a probabilistic finite-state machine, by assembling predefined behavioral modules that have been a priori defined in a mission-agnostic way. In the experiments presented in the paper, Gianduja and its variants were able to find solutions that revealed, at visual inspection, to properly leverage the communication capabilities of the robots by assigning a meaningful semantics to the messages. With a single exception, this holds true for all the missions we considered. When aggregated, the results we obtained show that Gianduja is the best method for one bit of information and Gianduja 2E is the best one for missions requiring more bits of information to be exchanged. Our results do not provide any conclusive evidence on which variant of Gianduja is the best one: further investigation is needed to understand the impact of their underlying design choices. Nonetheless, the results obtained in this article show that it is possible to automatically find solutions that leverage the use of communication in a swarm of robots controlled by probabilistic finite-state machines. Our contention is that by constraining the design space, it is easier for the optimization algorithm to find solutions where the listener’s and speaker’s behaviors emerge simultaneously. That being a requirement allowing communication to emerge. Future work will focus on studying the relationship between the complexity of a mission and the number of bits of information required to solve it. Our observations also corroborate the hypothesis that, introducing bias into the design phase of the control software in the form of pre-designed behavioral modules allows one to have better control on the design space and results in an increased robustness to the (pseudo-) reality gap.

Figure 7 Friedman test on the aggregate results of the three missions.

The plot represents the average rank of the six methods and their 95% confidence interval. The lower the better.

Table 4 Relative median performances of Gianduja {3,3E} and EvoCom 2 with respect to Gianduja 2 and Gianduja 2E.

	with respect to Gianduja 2	with respect to Gianduja 2E	
	Gianduja 3	EvoCom 2	Gianduja 3E	EvoCom 2	
BEACON AGGREGATION	−11.7%	−83.7%	−18.9%	−83.0%	
BEACON STOP	−12.9%	−25.7%	−1.06%	−35.5%	
BEACON DECISION	+7.44%	−2.89%	−3.10%	−5.23%	

Finally, although the results of the study we performed using a pseudo-reality indicate that Gianduja and its variants are relatively robust to the reality gap, further robots experiments should be performed to reliably characterize the performance of these methods.

The authors thank Frédéric Robert for contributing to the definition of the missions of Study A.

Additional Information and Declarations

Competing Interests

Author Contributions

Data Availability

1 The choice of performing a single evaluation of each instance of control software produced is dictated by statistical considerations. Given a fixed number of evaluations, this setting is the one that minimizes the variance of the estimation of the expected performance. For a discussion see Birattari (2020) and the references therein. For a formal proof—albeit refering to the different but formally equivalent problem of estimating the expected performance of a heuristic optimization algorithm—see Birattari (2004) and Birattari (2009).

Mauro Birattari is an Academic Editor for PeerJ.

Ken Hasselmann conceived and designed the experiments, performed the experiments, analyzed the data, performed the computation work, prepared figures and/or tables, authored or reviewed drafts of the paper, and approved the final draft.

Mauro Birattari conceived and designed the experiments, analyzed the data, authored or reviewed drafts of the paper, and approved the final draft.

The following information was supplied regarding data availability:

All supplementary data and code are available at IRIDIA - Supplementary Information ID IridiaSupp2019-005: http://iridia.ulb.ac.be/supp/IridiaSupp2019-005/.

Code is also available at GitHub: https://github.com/KenN7/argos3-AutoMoDe/.

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
