# Peer review of "Modular automatic design of collective behaviors for robots endowed with local communication capabilities"

_PeerJ Computer Science, doi:10.7717/peerj-cs.291_

## Round 0.1 · original submission · Minor Revisions

Both reviewers find the paper an exciting contribution to the collective behavior of the robot swarm research community. However, both reviewers have made a thorough and constructive review indicating different points where the article can be clarified or improved. In my opinion, addressing the issues raised by the reviewer will help the work.

·

Basic reporting

no comment

Experimental design

no comment

Validity of the findings

no comment

Additional comments

This paper describes a comparative study of an automatic design method for swarm robotics systems. In particular, it investigates the effect of message exchange on generating effective collective behaviours for robot swarms using a developed automatic designed method named Gianduja. The study describes different variants of Gianduja based on the increasing complexity of exchanged messages (i.e., by increasing number of bits in a message) then compares Gianduja and its variants with another automatic design method (e.g., EvoCom
and Chocolate).

Generally, this paper is properly organised and follows good and extensive experiments to provide sufficient conclusive evidence on the outcomes of this research. In my opinion, the results comprise useful message to the research community in the field of automatic design methods to generate collective behaviour of robot swarm.

The paper is well written with clear English language. To the best of my English, I could not spot any language problems in the paper (except for few missing spaces e.g., “the (pseudo-)reality gap. L659, P22”, “generated by Gianduja2E’performs”, L673, P22 ).

The paper provides elaborative supplementary materials which complement the material described in the paper. Furthermore, the Appendix section provides further experiments to explain unclear findings in the paper and to provide empirical evidence on the conjecture of the authors about the results.

However, I have few comments to be addressed in the followings:


- Why the white area in the Stop experiment (in study A) is smaller than the white and black areas in other experiments. Although it seems that in the presence of massage exchange, only a few robots need to find the white spot to signal the rest of robots to stop. However, the larger size of the white area:
i) increase the probability of finding the spot within the evaluation time (i.e., 120 s).
ii) can accommodate the majority of robots inside it.
This could ultimately guide the optimization process to relying on finding the spot rather than on receiving a signal from other robots. It seems the smaller size of the spot adds bias toward relying upon message exchange to increase the fitness of the group. Please, add a justification text in the paper about why the white area in the Stop experiment is smaller than the areas in other experiments in study A.

- Line 343, p9 “For Chocolate, Gianduja, and EvoCom, the control software produced is run once also on real robots”. This statement is unclear to me. Do you mean only one evaluation trial? If yes, why only one? If not, please, mention the number of evaluations conducted using real-robots. I could not find this information in the paper. This information is important to provide better evidence that validates your findings.

- The experiment runs on real robots need to be explained. How the parameters of the objective functions are measured (what tools are used to do that). For example, In the Decision experiment, how do you measure that the robots are in which half of the arena at the end of the evaluation (after 120 s elapses). Given that some robots in continuous movements. Similarly, in Stop experiment, how to measure the robot is moving or not at the end of every evaluation. If you did this measure by visual inspection of the recorded videos, please mentioned that in the paper.

- In Figure2, p11, the graphs do not show boxes for evaluations with real-robot for GiandujaE. Is that because you did these evaluations for Gianduja and since GiandujaE is derived from Gianduja and differs from it only in the way received messages are handled. If this is true, please, mentioned it in the paper.

- In probabilistic finite state machine graphs in figure3 and figure8, p12 and p18 respectively. Please indicates what are the symbols p, w, t, brd, rwm and att refer to. Please, do this in the figure caption or as a legend in the graph.

- In figure6, p16. It would be more clear if you used a different colour for the BECON robot to distinguished it from the rest of the group.

·

Basic reporting

The introduction provides useful background on the automatic design of collective behaviours for robot swarms, and sets the research in the context of previous work on AutoMoDe, clearly stating the aims of the article.

The related work section thoroughly reviews relevant literature on inter-robot communication, covering previous attempts to automatically design communication for multi-robot systems and robot swarms, strongly motivating the work presented.

The rest of the article is well-structured, and presents the research in a logical progression. The supplementary material is a valuable addition, particularly the videos of the generated behaviours.

Although the PFSM notation has been covered in previous work, Figures 3 and 8 would benefit from a legend to explain the symbols used, to make this article self-contained.

There are a few grammatical errors and typos throughout - please proofread carefully before final submission.

Experimental design

The experimental design is strong - the swarm is sufficiently large, standard tools (e-puck robots and ARGoS3 simulator) are used, appropriate case study swarm tasks were chosen, and the performance of Gianduja (and variants) is compared against that of AutoMoDe-Chocolate and EvoCom.

The data gathering is rigorous - experiments are carried out in simulation, "pseudo-reality", and on real robots. The methods are described in sufficient detail to be replicated, and source code is available online in a GitHub repository.

It is unclear why GiandujaE was not tested on real robots in Study A - only in simulation and pseudo-reality. The authors state that "After having validated the ability of pseudo-reality to provide reliable evaluations, we will adopt it as the exclusive device to perform the experiments on Study B. Performing these experiments with real robots would be time consuming and would not give any further contribution to the scientific message of the paper". However, the results shown in Figure 2 are part of Study A, so I expected to see the results of real-robot trials for GiandujaE here.

Validity of the findings

There is sufficient replication of each experimental treatment, and non-parametric statistical testing is used appropriately to validate the findings. The conclusions drawn are consistent with the results presented.

I could not find the raw data that the box plots were generated from though - will this be included in the supplementary material?

There are clear differences in the performance between the pseudo-reality and real-robot experiments for Study A (Figure 2). Carrying out real-robot assessment of GiandujaE in Study A, and for all treatments in Study B would strengthen the validity of the findings. However, if this is infeasible, could you comment further on the expected performance on real robots?

Additional comments

This research into the automatic design of communication is a valuable contribution to the field of swarm robotics, and provides critical insight into the effect of increasing the amount of information exchanged between robots.

---

## Round 0.2 · accepted · Accept

The reviewed paper includes all the relevant suggestions and comments previously addressed by the reviewers